# Deep Learning in Gynecologic Cancer Diagnosis: Current Advances, Challenges, and Future Directions

## Abstract

Gynecologic cancers remain a significant health challenge globally, particularly in low-resource settings where diagnostic consistency and expertise are often limited. Traditional diagnostic methods based on human image analysis can be inconsistent and prone to errors. Recent advancements in deep learning (DL) offer promising solutions for automating image analysis, providing more objective and accurate diagnostic outcomes. This review synthesizes current advances, identifies challenges, and explores future directions in the application of DL for gynecologic cancer diagnosis using various imaging modalities. A thorough literature review following PRISMA-2 guidelines examined studies employing DL to diagnose various gynecologic cancers through imaging modalities like MRI, CT scans, Pap smears, and colposcopy. Data extraction and quality assessment were conducted using the QUADAS-2 tool, and diagnostic performance was evaluated through pooled sensitivity, specificity, SROC curves, and AUC metrics using R software. From 48 studies reviewed, 24 met the inclusion criteria for the meta-analysis. The studies employed various DL models, predominantly ResNet, VGGNet, and UNet, across different imaging modalities. DL models demonstrated superior sensitivity (89.40%) but slightly lower specificity (87.6%) compared to traditional machine learning (ML) methods, which exhibited 68.1% sensitivity and 94.1% specificity. The AUC for DL models was 0.88, underscoring their high diagnostic accuracy. Challenges such as study heterogeneity and methodological biases were identified, underscoring the importance of standardized protocols. Despite these obstacles, DL holds significant promise in gynecologic cancer diagnosis, particularly in resource-constrained settings. Addressing these challenges can enhance the clinical utility of DL and contribute to improved patient outcomes.

## 1 Introduction

Gynecologic cancers, including cervical, ovarian, and endometrial cancers, pose significant health risks to women globally, especially in low-resource settings[11, 2, 3]. Traditional diagnostic methods relying on human interpretation are inconsistent and error-prone [2, 3, 7]. Recent advances in DL offer automated, consistent, and precise diagnostic capabilities using medical images [5, 10, 1, 12, 8, 4, 6, 13]. This paper reviews recent DL applications in gynecologic cancer diagnosis, discusses their performance compared to conventional methods, and highlights future research directions aimed at overcoming current challenges.

## 2  Methods

A systematic review and meta-analysis following PRISMA-2[9] guidelines were conducted. The protocol was registered in PROSPERO under registration number 356104. We searched databases including PubMed, Embase, and Scopus for articles published between January 2018 and December 2022, focusing on DL applications for diagnosing several gynecologic cancers. Articles were screened and assessed using QUADAS-2 [14] for eligibility and quality. Data on diagnostic performance metrics, specifically sensitivity and specificity, were extracted and analyzed using the "meta" and "mada" packages. Pooled sensitivity, specificity, summary receiver operating characteristic (SROC) curves, and area under the curve (AUC) metrics were calculated using R software.

## 3  Experimental Results

The review included 48 studies, with 24 studies eligible for meta-analysis. The imaging modalities used across these studies included cytology (20 studies), colposcopy (15 studies), MRI (8 studies), CT scans (4 studies), and hysteroscopy (1 study). Popular DL models were ResNet, VGGNet, and UNet.

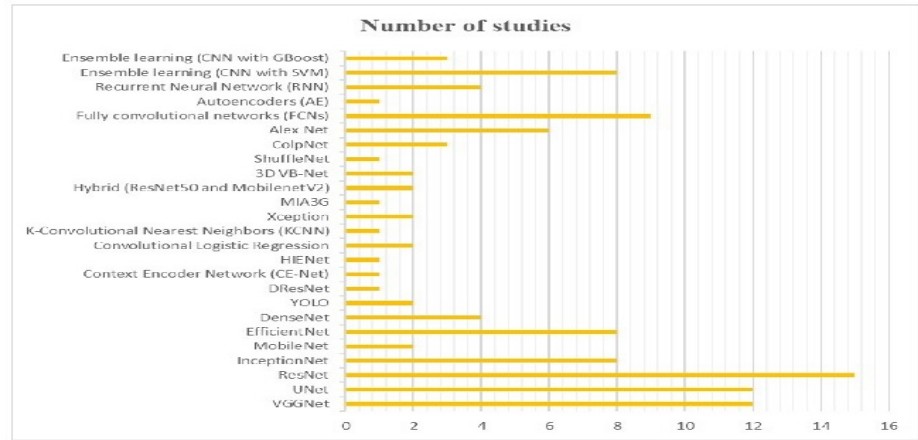

Figure 1: Overview of the diversity and popularity of models used in different studies for the diagnosis of gynecologic cancers

DL algorithms demonstrated high sensitivity (89.40%) but lower specificity (87.6%) compared to ML methods, which had lower sensitivity (68.1%) and higher specificity (94.1%). The AUC for DL algorithms was 0.88, indicating good diagnostic accuracy. 3D-UNet and EfficientNet-B3 were top-performing models, with EfficientNet-B3 achieving a classification accuracy of 99.01% for cervical cancer detection and 3D-UNet demonstrating high segmentation capabilities with an average Dice Similarity Coefficient (DSC) score of 0.93. Despite the promising results, significant heterogeneity and risk of bias were noted, primarily due to variability in patient selection, imaging techniques, and DL model implementation.

## 4  Conclusion

Deep learning techniques offer substantial improvements in the accuracy and efficiency of diagnosing gynecological cancers using image-based data, surpassing traditional ML methods. However, challenges related to study heterogeneity and model biases need addressing to enhance the clinical applicability of DL models. Future research should focus on refining these algorithms, ensuring their robustness and generalizability, and addressing ethical considerations to fully leverage the potential of DL in gynecological cancer care. Collaborations between researchers, clinicians, and technologists are essential for advancing DL applications in gynecologic cancer diagnostics, ultimately improving early detection and patient outcomes.

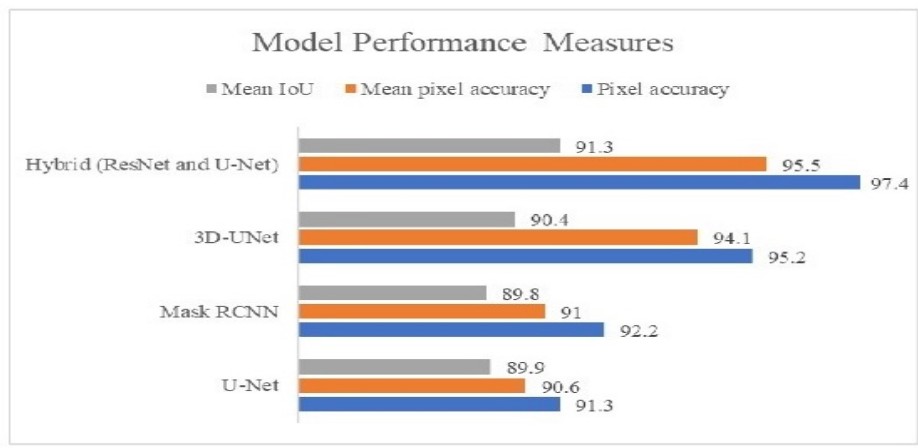

Figure 2: A Deep learning models used for abnormality detection

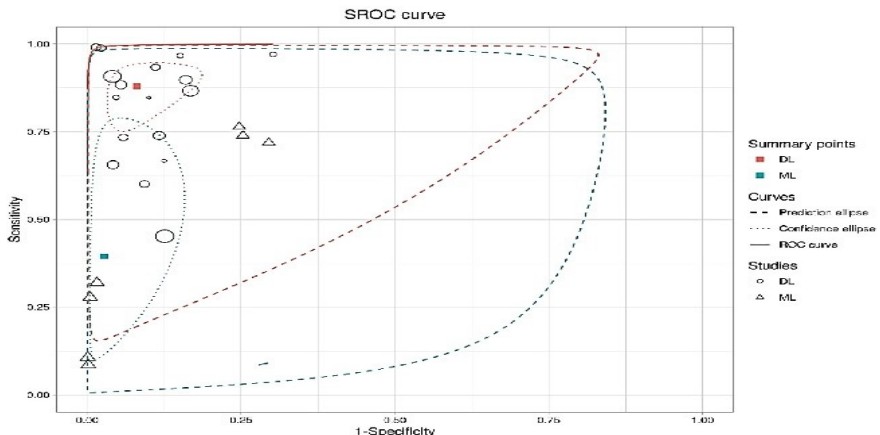

Figure 3: Summary of the receiver operating characteristic (SROC) plot of the advanced machine learning algorithms

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
