# OpenReview forum: "Deep Learning in Gynecologic Cancer Diagnosis:  Current Advances, Challenges, and Future Directions"
_IEEE.org/ICIST/2024/Conference — IEEE ICIST 2024 Conference Submission_

### Official Review · Reviewer_vbfo · 2024-08-22
**This article is acceptable after detailed revisions**

**Rating:** 6
**Confidence:** 3

**Review:**

This review synthesizes current advances, identifies challenges, and explores future directions in the application of DL for gynecologic cancer diagnosis using various imaging modalities. The obtained result is valuable and can be accepted if the following problems can be clarified.
(1) In the introduction, the shortages of those relevant studies are suggested to be further summarized.
(2) There exist several spelling and grammar errors. Please check carefully and further polish
 (3) In the Methods and Experimental results, more analysis can be added to better explain the main contents of this paper.
 (4) The future work is missing in the Conclusion.
(5) The references should be updated and their format standardized for enhanced consistency and accuracy.

---

### Official Review · Reviewer_rvsK · 2024-08-22
**This paper investigated employing DL to diagnose 9 various gynecologic cancers through imaging modalities like MRI, CT scans, Pap 10 smears, and colposcopy. The topic of this paper is interesting.**

**Rating:** 8
**Confidence:** 4

**Review:**

Comments to the Author
This paper investigated employing DL to diagnose 9 various gynecologic cancers through imaging modalities like MRI, CT scans, Pap 10 smears, and colposcopy. The topic of this paper is interesting. Below is a list of comments that should be taken into account further when revising the paper.
1.	In the introduction section, the order of references should be from small to large, which will make the overall structure of the article more beautiful.
2.	In the methodology section, the adopted method should be explained to make readers more aware of why it is used and its benefits.
3.	The paper should provide a detailed description of the innovative points to enable readers to quickly understand the article. Meanwhile, please elaborate on the future plans.

---

### Official Review · Reviewer_h5Zj · 2024-08-23
**accept**

**Rating:** 7
**Confidence:** 3

**Review:**

This paper synthesizes current advances, identifies challenges, and explores future directions in the application of DL for gynecologic cancer diagnosis using various imaging modalities. The theory is correct and can be accepted after responding the following comments.
(1) There are many typos and grammar errors. The authors should have a native English speaker or software packages to perform the editing check.
(2)What is the contribution of the paper? It should be highlighted both in the introduction and in the content.
(3)The conclusion suggests adding a section on the prospects for future research.

---

### Comment · Reviewer_vbfo · 2024-08-21
**This article is acceptable after detailed revisions**

This review synthesizes current advances, identifies challenges, and explores future directions in the application of DL for gynecologic cancer diagnosis using various imaging modalities. The obtained result is valuable and can be accepted if the following problems can be clarified.
(1)	In the introduction, the shortages of those relevant studies are suggested to be further summarized.
(2)	There exist several spelling and grammar errors. Please check carefully and further polish
(3)	In the Methods and Experimental results, more analysis can be added to better explain the main contents of this paper.
(4)	The future work is missing in the Conclusion.
(5)	The references should be updated and their format standardized for enhanced consistency and accuracy.

---

### Decision · Program_Chairs · 2024-09-06

Accept (Oral)